Recent population expansion of longtail tuna Thunnus tonggol (Bleeker, 1851) inferred from the mitochondrial DNA markers

Syahida Kasim Noorhani 1
Mat Jaafar Tun Nurul Aimi 1
Mat Piah Rumeaida 1
Mohd Arshaad Wahidah 2
Mohd Nor Siti Azizah 3
http://orcid.org/0000-0003-1702-6111 Habib Ahasan 1 4
Abd. Ghaffar Mazlan 3
Sung Yeong Yik 3
http://orcid.org/0000-0001-9438-0128 Danish-Daniel Muhd 1 3
http://orcid.org/0000-0003-0149-5247 Tan Min Pau 1 3 mptan@umt.edu.my
1 Faculty of Fisheries and Food Science, Universiti Malaysia Terengganu , Terengganu , Malaysia
2 Marine Fishery Resources Development and Management Department (MFRDMD) , Taman Perikanan, Chendering, Kuala Terengganu , Malaysia
3 Institute Marine Biotechnology (IMB), Universiti Malaysia Terengganu , Terengganu , Malaysia
4 Department of Fisheries and Marine Science, Noakhali Science and Technology University , Noakhali , Bangladesh
Rahman Mohammad Shamsur
Electronic publication date: 2020 Aug 6
Publication date: 2020
Volume: 8
Electronic Location ID: e9679
Received 2019 Aug 15; Accepted 2020 Jul 17
Copyright: © 2020 Syahida Kasim et al.
Copyright year: 2020
Copyright holder: Syahida Kasim et al.
License: This is an open access article distributed under the terms of the Creative Commons Attribution License, which permits unrestricted use, distribution, reproduction and adaptation in any medium and for any purpose provided that it is properly attributed. For attribution, the original author(s), title, publication source (PeerJ) and either DOI or URL of the article must be cited.
License URL: https://creativecommons.org/licenses/by/4.0/

Keywords: Genetic diversity, Mitochondrial DNA, Control region (D-loop), Thunnus tonggol, NADH dehydrogenase subunit 5 (ND5), Population expansion, Longtail tuna

Funding: Ministry of Higher Education (MOHE) FRGS/1/2016/WAB13/UMT/03/1 This work was financially supported by the Ministry of Higher Education (MOHE), FRGS/1/2016/WAB13/UMT/03/1. The funders had no role in study design, data collection and analysis, decision to publish, or preparation of the manuscript.

==============================
The population genetic diversity and demographic history of the longtail tuna Thunnus tonggol in Malaysian waters was investigated using mitochondrial DNA D-loop and NADH dehydrogenase subunit 5 (ND5). A total of 203 (D-loop) and 208 (ND5) individuals of T. tonggol were sampled from 11 localities around the Malaysian coastal waters. Low genetic differentiation between populations was found, possibly due to the past demographic history, dispersal potential during egg and larval stages, seasonal migration in adults, and lack of geographical barriers. The gene trees, constructed based on the maximum likelihood method, revealed a single panmictic population with unsupported internal clades, indicating an absence of structure among the populations studied. Analysis on population pairwise comparison ФST suggested the absence of limited gene flow among study sites. Taken all together, high haplotype diversity (D-loop = 0.989–1.000; ND5 = 0.848–0.965), coupled with a low level of nucleotide diversity (D-loop = 0.019–0.025; ND5 = 0.0017–0.003), “star-like” haplotype network, and unimodal mismatch distribution, suggests a recent population expansion for populations of T. tonggol in Malaysia. Furthermore, neutrality and goodness of fit tests supported the signature of a relatively recent population expansion during the Pleistocene epoch. To provide additional insight into the phylogeographic pattern of the species within the Indo-Pacific Ocean, we included haplotypes from GenBank and a few samples from Taiwan. Preliminary analyses suggest a more complex genetic demarcation of the species than an explicit Indian Ocean versus Pacific Ocean delineation.

Introduction

Thunnus tonggol, locally known as aya/tongkol hitam or longtail tuna, is a pelagic-neritic marine fish classified in the subgenus Neothunnus from the tribe Thunini within family Scombridae along with blackfin (T. atlanticus) and yellowfin (T. albacares) tuna (Chow & Kishino, 1995). It is the second smallest Thunnus species (Griffiths et al., 2019) but is also reported as the largest growing species among neritic tuna (Koya et al., 2018). T. tonggol is distributed exclusively in the Indo-Pacific region between 47°N and 33°S (Froese & Pauly, 2011) and is one of the most economically important species in Southeast Asia (Itoh, Tsuji & Chow, 1996; Hidayat & Noegroho, 2018). It represents essential, artisanal, and sustenance fisheries as one of the biggest sources of wild caught food (Frimodt, 1995; Collette, 2001). It is also considered an important sport-fish due to its large size and fighting ability (Griffiths et al., 2019).

The annual global catch of T. tonggol has tripled in the last 20 years (Griffiths et al., 2019). As a consequence, there was a marked decrease in its catch from 291,264 tonnes to 237,124 tonnes (FAO, 2018) from 2007 to 2016. The highest catches reported were from Iran, Indonesia, Pakistan, Malaysia, Oman, Yemen, India, and Thailand (Pierre, Geehan & Herrera, 2014; Koya et al., 2018), where they were mainly caught using gillnet fleets operating in coastal waters. In Malaysia, the landing of neritic tuna consists of longtail tuna (T. tonggol), kawakawa (Euthynnus affinis), and frigate tuna (Auxis thazard and A. rochei) and contributed 5% of the total marine landings (Mohd Faizal et al., 2019) with T. tonggol dominating, followed by kawakawa (E. affinis), and frigate tuna (A. thazard and A. rochei) (Samsudin et al., 2018).

Basic population parameters, such as the number and distribution of stocks, as well as population genetic diversity are very much needed for a sound management program. For example, population stock data is essential to support resource recovery and to aid in delineating and monitoring populations for fishery management (Roldan et al., 2000; Kunal et al., 2014). Despite its important contributions, coupled with steadily increasing demands in recent years, there is little research on T. tonggol; it has, in fact, received less attention than other pelagic species in Southeast Asian waters (Willette, Santos & Leadbitter, 2016), including Malaysia. A previous study on population genetics of T. tonggol from the northwest coast of India, based on the mitochondrial displacement loop (D-loop) marker, revealed low genetic differentiation between localities, which suggested a panmictic stock structure (Kunal et al., 2014). However, a study across a wider spatial coverage, also based on the D-loop marker, suggested geographical segregation for T. tonggol within the Indo-Pacific region (Willette, Santos & Leadbitter, 2016). In other tuna species, such as the bigeye tuna (T. obesus), populations from the South China Sea, the Philippines Sea, and western Pacific Ocean consist of a single intermixing identity (Chiang et al., 2006), while yellowfin tuna (T. albacore) along the Indian coast, displayed multiple geographically distinct stocks (Kunal et al., 2013).

Population genetics is an essential tool to improve knowledge on stock delineation and population dynamics of exploited fish (Hauser & Seeb, 2008). Genetic markers, like mitochondrial DNA (mtDNA), has proven to be one of the most efficient tools for evaluating intraspecific genetic variation and to describe population genetics study (Menezes, Kumar & Kunal, 2012; Tan, Jamsari & Siti Azizah, 2012; Nabilsyafiq et al., 2019). Moreover, it is also widely used in evolutionary genetics as markers for population history and to estimate divergence times among taxa (Tan et al., 2020). MtDNA is considered a sensitive and reliable marker (Hoolihan, Anandh & Herwerden, 2006; Habib & Sulaiman, 2016, 2017; Tan et al., 2019) due to its large quantity in the cell and elevated mutation rate (11.6 times higher than nuclear DNA (Allio et al., 2017)), a consequence of a non-existent or inefficient repair system. In addition, the lack of recombination in mtDNA, coupled with relatively infrequent gene arrangement, makes it a good choice for population genetics study (De Mandal et al., 2014).

The mtDNA NADH dehydrogenase subunit 5 (ND5) and the non-coding and highly polymorphic D-loop markers were adopted to conduct the first population genetics and phylogeographic studies among wild populations of T. tonggol in Malaysian coastal areas. The utilization of D-loop as a population genetic marker has been widely documented in a plethora of marine species, including in tuna (Durand et al., 2005; Chiang et al., 2006; Carpenter et al., 2011). However, mtDNA ND5 gene is rarely used in marine species. Nevertheless, this protein-coding gene contains both slow and rapid evolving regions that permit its application in elucidating the genetic relationships among populations; for instance, six populations of the Persian sturgeon Acipenser persicus from the south Caspian Sea were found to be genetically differentiated, as inferred from the ND5 polymerase chain reaction-restriction fragment length polymorphism (PCR-RFLP) assay (Pourkazemi et al., 2012); meanwhile, moderate genetic differentiation among populations of masu salmon Oncorhynchus masou masou from Japan, Russia, and Korea was apparent based on ND5 and microsatellites markers (Yu et al., 2010). However, low genetic differentiation with high gene flow was detected between sampling locations of the Pearse’s mudskipper (Periophthalmus novemradiatus) that inhabits Setiu Wetland, Terengganu, Malaysia (Nabilsyafiq et al., 2019).

The present study aimed at elucidating the population genetics of T. tonggol in Malaysian waters based on mtDNA D-loop and ND5 markers, as well as further investigating the phylogeography of T. tonggol within the Indo-Pacific region, where samples from Taiwan (provided by National Taiwan University) and India (west coast: Kochi, Veraval, and Ratnagiri, and east coast: Andaman Sea; all haplotype sequences retrieved from GenBank) were included in molecular analyses. The output from this study would be beneficial for the conservation and management of this species.

Materials and Method

Ethical statement

Only a small clipping of the pectoral fin from each individual fish was collected from the local wet markets. This species is not in the IUCN list of endangered or protected species. As only dead specimens were sampled, no permit was required and no ethical consideration was linked to the study.

Sample collection

The specimens were collected from 11 fish landing sites in Malaysia and were morphologically identified following identification keys as described in Lim et al. (2018). This species can be differentiated from other Thunnus species by having a moderate length pectoral fin, reaching the origin of the second dorsal fin and blackish caudal fin (Lim et al., 2018). A small portion of pectoral fin of each individual was cut and preserved in 95% ethanol and stored in a 1.5 mL centrifuge tube at 4–8 °C until further analysis. Each catch locality was confirmed to be non-overlapping (discrete geographical entity) based on feedback from the fishermen and was divided into four regions following Akib et al. (2015) (Table 1; Fig. 1A). Additionally, four samples from Taiwan and 153 GenBank sequences of T. tonggol from the Indian waters (Fig. 1B) were included for phylogeographic analysis.

Figure 1 Sampling locations of Thunnus tonggol along the Malaysian coastal waters (A) and additional samples (B). (Modified from source: http://www.supercoloring.com/coloring-pages/malaysia-map).

KP, Kuala Perlis; PR, Pantai Remis; SB, Sungai Besar; TG, Pulau Kambing; KT, Kuantan; TB, Tok Bali; KC, Kuching, BT, Bintulu; MR, Miri; KK, Kota Kinabalu; SM, Semporna; TW, Taiwan; KH, Kochi; VR, Veraval; RN, Ratnagiri; AD, Andaman Sea. Exact coordinates of additional samples are unknown.

Table 1 Sampling locations, coordinates and collection date of 11 Thunnus tonggol populations from the surrounding seas (in the Indo Pacific region) of Malaysia.

Region	No	Population	Latitude (North)	Longitude (East)	Sampling date	
Strait of Malacca (SOM)	1	Kuala Perlis (KP), Perlis	6°23′55.4″N	100°07′43.4″E	19/07/2017	
2	Pantai Remis (PR), Perak	4°31′42.0″N	100°38′39.0″E	17/05/2018	
3	Sungai Besar (SB), Selangor	3°39′50.4″N	100°59′16.6″E	15/05/2018	
South China Sea 1 (SCS-1)	4	Pulau Kambing (TG), Terengganu	5°19′19.7″N	103°07′42.3″E	02/10/2017	
5	Kuantan (KT), Pahang	3°47′14.9″N	103°19′04.7″E	19/07/2017	
6	Tok Bali (TB), Kelantan	5°52′35.5″N	102°27′29.9″E	10/07/2017	
South China Sea 2 (SCS-2)	7	Kuching (KC), Sarawak	1°33′27.0″N	110°21′38.7″E	10/03/2018	
8	Bintulu (BT), Sarawak	3°10′13.8″N	113°02′25.8″E	08/03/2018	
9	Miri (MR), Sarawak	4°23′31.1″N	113°59′07.6″E	07/03/2018	
10	Kota Kinabalu (KK), Sabah	5°58′59.4″N	116°04′22.5″E	19/07/2017	
Celebes Sea (CS)	11	Semporna (SM), Sabah	4°28′49.8″N	118°36′39.7″E	20/03/2018	

Genomic DNA isolation and polymerase chain reaction amplification

Genomic DNA was isolated from fin tissue by using the salt extraction method (Miller, Dykes & Polesky, 1988). The isolated DNA samples were then PCR amplified with the mtDNA partial displacement loop (D-loop) and ND5. The following primers were used: (1) D-loop—Pro889U20 (5′-CCW CTA ACT CCC AAA GCT AG-3′, forward) and TDKD1291L21 (5′-CCT GAA ATA GGA ACC AAA TGC-3′, reverse) (Sulaiman & Ovenden, 2009) (2) ND5—L12321-Leu (5′-GGTCTTAGGAACCCAAAACTCTTGCTGCAA-3′, forward) and H13396-ND5 (5′-CCTATTTTKCGGATGTCYTG-3′, reverse) (Ruzainah, 2008). The PCR reaction mixture consisted of 2 µl of the DNA template, 0.5 µl of each primer, 12.5 µl of MyTaq™ Red Mix (Bioline), and 9.5 µl sterilized ultrapure water (ddH2O) with a final volume in each tube of 25 µl. The temperature profile for the D-loop was: initial denaturation at 94 °C for 5 min followed by 32 cycles of 94 °C for 30 s, 55 °C for 30 s, 72 °C for 1 min, final extension at 72 °C for 5 min, and final hold at 4 °C. Amplification conditions for the ND5 gene were: initial denaturation at 94 °C for 2 min followed by 35 cycles for 94 °C for 20 s, 59 °C for 20 s, 72 °C for 1 min 30 s, final extension at 72 °C for 5 min, and final hold at 10 °C. PCR products were visualized on 1.7% agarose gels stained with SYBR Safe to confirm their presence and estimate the size of DNA fragment amplified. PCR products were then sent for sequencing (First BASE Laboratories Sdn Bhd, Selangor, Malaysia) in the forward direction only using an Applied Biosystem ABI3730x1 capillary-based DNA sequencer.

Sequence editing and alignment

Multiple sequences were aligned and edited using ClustalW implemented in MEGA 6.0 (Tamura et al., 2013). DNA sequences were verified for correct identity by using the Basic Local Alignment Search Tool (BLAST) in the National Center for Biotechnology Information (NCBI) database (http://blast.ncbi.nlm.nih.gov/Blast.cgi) before further analyses. All haplotypes were deposited into GenBank under the accession numbers MK643829–MK644008 (D-loop) and MN252922–MN252981 (ND5).

Data analyses

Genetic diversity

The complete aligned datasets were used to estimate the number of haplotypes, haplotype diversity (H), and nucleotide diversity (π) in DnaSP 6.0 (Rozas et al., 2017). The polymorphic and parsimony informative sites were examined in MEGA 6.0.

Phylogenetic and population level analyses

The phylogenetic relationships among haplotypes were determined based on the maximum likelihood (ML) method implemented in MEGA 6.0. The best nucleotide substitution models with the lowest BIC score (Bayesian Information Criterion) for the D-loop and ND5 sequences were Tamura 3-parameter (T92) (Tamura, 1992) and Hasegawa–Kishino–Yano with Gamma distribution and invariant sites (HKY+G+1), respectively, as identified in MEGA 6.0. In the case that the T92 and HKY models were unavailable in the BEAST and Arlequin software packages (see below), the TN93 model (Tamura & Nei, 1993) was used instead. The confidence level for each node was assessed by 1,000 bootstrap replications (Felsenstein, 1985). The Pacific bluefin tuna T. orientalis (AB933631) was included as an out-group taxon for D-loop sequences, while the Yellowfin tuna, T. albacares (KM588080) was included as out-group taxon for ND5 gene sequences. To infer the relationships among haplotypes from Malaysian waters, a minimum spanning network (MSN) was constructed by using the median-joining method implemented in NETWORK version 5.0.1.1 (Bandelt, Forster & Rohl, 1999).

The population pairwise comparisons ФST for both datasets were determined using Arlequin 3.5 software (Excoffier & Lischer, 2010) and the statistically significant pairwise comparisons were tested with 10,000 permutations. Significant probability values were corrected by performing the False Discovery Rate Procedure (FDR) at α = 0.05 (Benjamini & Hochberg, 1995). Further analysis of genetic differentiation among populations was extended for haplotype-based statistics (HST), sequence-based statistics (NST) (Lynch & Crease, 1990), and KST* with significance levels assessed using permutation tests with 1,000 replicates (Hudson, Boos & Kaplan, 1992) in DnaSP 6.0. Using the same program, the estimation of gene flow (Nm) based on both haplotype and sequence statistics were calculated following Nei (1973) and Hudson, Boos & Kaplan (1992), respectively. Genetic distance estimates between sampled populations were calculated in MEGA 6.0.

Analysis of molecular variance (AMOVA) was performed to infer the population subdivision with three hierarchical levels, including genes within individuals, individuals within demes, and demes within groups of demes (Excoffier, Laval & Schneider, 2005), by using Arlequin 3.5 software. The Mantel test in IBD v 1.52 (Isolation by Distance) (Mantel, 1967; Bohonak, 2002) was used to investigate the correlation between genetic and geographical distance. Genetic distance was represented by population pairwise ФST values while geographical distances between sampling locations were measured by using Google Earth. Geographic distance was ln transformed and the strength of the relationship was examined with reduced major axis regression (10,000 randomizations) in IBD v1.52.

To understand its phylogeography within the Indo-Pacific region, GenBank sequences of the D-loop obtained from the west coast of the Indian Ocean (WCIO) (Kochi (25 haplotypes) MF592988–MF593012, Veraval (22) MF593027–MF593048, Ratnagiri and Veraval (92) KC313300–KC313393 (Kunal et al., 2014)), east coast of the Indian Ocean (ECIO) (Andaman Sea (14) MF593013–MF593026), and four samples from Taiwan (TW) were included in phylogenetic tree reconstruction, pairwise comparisons ФST, AMOVA, and genetic distance analyses. Altogether 309 haplotype sequences were realigned, and ambiguous GenBank sequences were eliminated. Only 152 Malaysian haplotypes (out of 180 haplotypes, see “Results”) were used after trimming the sequence to a final length of 388 base pairs (bp).

Demographic history

Historical demographic and spatial expansions were inspected in the T. tonggol populations. Fu’s FS (Fu, 1997) and Tajima’s D (Tajima, 1989) were adopted to analyze deviation from neutrality. Historical demographic parameters, including the population before expansion (ϴ0), after expansion (ϴ1), and relative time since population expansion (τ), were computed in Arlequin 3.5. The values of time (τ) were transformed to estimate the actual time (T) since population expansion, using the equation τ = 2μt, where t is the age of the population in generations and µ is the sequence mutation rate per generation. In the present study, a mutation rate of 3.6 × 10−8 mutation per site/year was applied for the D-loop (Donaldson & Wilson, 1999) and 2% per million years for the ND5 (Brown, George & Wilson, 1979). Bayesian skyline analyses were plotted using BEAST version 2.2.0 (Bouckaert et al., 2019), where the changes in effective population size (Ne) over time were tested. This enabled past demographic changes of T. tonggol to be inferred from the current patterns of genetic diversity within a population (Drummond et al., 2005). Since there was the absence of a population structure (see “Results”), a single population was modeled. The input was prepared in BEAUti. The analysis was run for 108 iterations with a burn-in of 107 with sampling every 104 and a strict molecular clock. All operators were automatically optimized and the results were generated using Tracer version 1.7.1 (Rambaut et al., 2018).

Harpending’s (1994) raggedness index (Hri) and sum of squared deviations (SSD) were computed in Arlequin 3.5 to evaluate if the sequence data significantly diverged from the assumptions of a population expansion model. The raggedness index has been shown to be a powerful tool in quantifying population growth with limited sample sizes (Ramos-Onsins & Rozas, 2002). In addition, the mismatch distribution was calculated in DnaSP 6.0. The pattern could be used to provide an insight of the past population demography (Chen et al., 2015). A population that has undergone recent expansion shows a unimodal distribution pattern, while a population in equilibrium shows a multimodal distribution pattern (Slatkin & Hudson, 1991; Rogers & Harpending, 1992).

Results

Genetic diversity

A total of 203 and 208 individuals were sequenced for the partial mtDNA D-loop and ND5 gene, respectively. The final alignment of D-loop sequences (416 base pairs (bp)) revealed 113 polymorphic sites (42 singletons and 71 parsimony informative sites), defining 180 haplotypes, where 14 (7.78%) were found in two to five localities and the rest (92.2%) were either private to a single locality or singleton haplotype. The ND5 sequences (855 bp) revealed 60 variable sites (34 parsimony informative sites, 26 singletons), defining 60 haplotypes, where 17 (27.9%) were shared by two to 11 populations, five (8.2%) were private to a single locality and 38 (63.9%) were singleton haplotypes. The D-loop region was AT rich, while, ND5 gene sequences demonstrated higher percentages of CG (56%). All populations of T. tonggol from Malaysian waters showed high haplotype diversity (D-loop: 0.990–1.000; ND5: 0.848–0.965) but low to moderate nucleotide diversity (D-loop: 0.0195–0.0250; ND5: 0.0017–0.0039) (Table 2).

Table 2 Molecular diversity, neutrality test, mismatch distribution and goodness of fit tests for Thunnus tonggol populations based on D-loop and ND5 sequences.

Population	Genetic diversity	Neutrality test	Mismatch distribution	Goodness of fit tests	
N	h (S)	H	π	Tajima’s D	Fu’s FS	ϴ0	ϴ1	τ	SSD	Hri	
D-loop	
KP	20	18 (20)	0.990	0.0239	−0.99	−7.32*	0	99,999.000	10.223	0.0072	0.0149	
PR	21	20 (46)	0.995	0.0238	−0.95	−10.66*	0.016	93.860	9.291	0.0012	0.0069	
SB	18	18 (45)	1.000	0.0250	−0.89	−10.13*	1.248	348.750	10.313	0.0034	0.0123	
TG	16	16 (33)	1.000	0.0200	−0.74	−9.66*	0.002	99,999.000	8.705	0.0093	0.0211	
KT	12	12 (27)	1.000	0.0195	−0.49	−5.88*	0.000	99,999.000	8.840	0.0114	0.0326	
TB	19	19 (43)	1.000	0.0245	−0.76	−11.34*	0.000	99,999.000	10.440	0.0093	0.0322	
KC	14	14 (37)	1.000	0.0238	−0.70	−6.70*	0.000	3,235.000	9.315	0.0107	0.0258	
BT	18	17 (36)	0.994	0.0203	−0.83	−8.76*	0.000	58.750	9.113	0.0125	0.0212	
MR	19	19 (44)	1.000	0.0236	−0.95	−11.64*	0.000	99,999.000	9.955	0.0048	0.0142	
KK	22	21 (52)	0.996	0.0238	−1.25	−11.68*	0.000	99,999.000	9.871	0.0096	0.0164	
SM	24	23 (43)	0.996	0.0216	−0.89	−14.95*	0.005	337.656	10.424	0.0032	0.0128	
Overall	203	180 (105)	–	–	–	–	–	–	8.514	–	–	
Mean	18	–	0.997	0.0227	−0.86	−9.88	0.116	54,915.300	9.681	0.0075	0.0191	
ND5		
KP	20	13 (14)	0.937	0.0022	−1.88*	−10.03*	0.000	99,999.000	2.039	2.0390	0.0295	
PR	19	14 (22)	0.965	0.0039	−1.85*	−8.34*	0.000	99,999.000	3.439	3.4394	0.0023	
SB	21	11 (14)	0.868	0.0022	−1.78*	−5.87*	0.000	99,999.000	1.933	1.9335	0.0012	
TG	16	10 (14)	0.917	0.0030	−1.52	−4.49*	0.000	99,999.000	1.439	1.4395	0.0465	
KT	13	10 (10)	0.923	0.0021	−1.80*	−7.93*	0.000	99,999.000	1.947	1.9473	0.0197	
TB	19	9 (9)	0.848	0.0017	−1.48	−4.85*	0.000	99,999.000	1.535	1.5352	0.0148	
KC	18	11 (14)	0.909	0.0028	−1.55*	−5.66*	0.000	15.340	2.748	2.7481	0.0304	
BT	18	12 (15)	0.941	0.0031	−1.34*	−6.34*	0.000	99,999.000	3.119	3.1191	0.0065	
MR	20	12 (18)	0.905	0.0030	−1.84*	−6.08*	1.130	216.470	1.650	1.6504	0.0009	
KK	21	12 (15)	0.929	0.0026	−1.71*	−6.74*	0.000	99,999.000	2.281	2.2813	0.0099	
SM	23	12 (15)	0.905	0.0027	−1.63*	−7.15*	0.000	99,999.000	2.488	2.4883	0.0023	
Overall	208	60 (60)	–	–	–	–	–	–	2.321	–	–	
Mean	19	–	0.916	0.0027	−1.67*	−6.68*	0.103	81,838.400	2.238	0.0149	0.0749	
Notes:

* Significant at P < 0.05.

N, number of individuals; h, number of haplotype; S, number of segregating sites; H, haplotypes diversity; π, nucleotide diversity; ϴ0/ϴ1: before/after expansion; τ, relative time since population expansion; SSD, sum of squared deviations; Hri, Harpending’s raggedness index; KP, Kuala Perlis; PR, Pantai Remis; SB, Sungai Besar; PK, Pulau Kambing; KT, Kuantan; TB, Tok Bali; KC, Kuching; BT, Bintulu; MR, Miri; KK, Kota Kinabalu; SM, Semporna.

Phylogenetic and population level analyses

The phylogenetic reconstruction inferred from the mtDNA D-loop region and ND5 gene revealed a gene tree with mainly unsupported clades (<50%) and obscure patterns of geographical segregation associated with genetic distribution (Fig. 2; Supplemental 1). This was aligned with the MSN haplotype network that showed no geographical partitioning among the populations studied. Specifically, 180 D-loop haplotypes showed a complex reticulated network (Fig. 3), while 60 ND5 haplotypes revealed a more clarified network pattern (Fig. 4). No dominant haplotype was detected based on the D-loop marker, however, Hap004 and Hap005 were considered the most abundant and common haplotypes, followed by Hap036 and Hap116. Among ND5 haplotypes, Hap01 was the most dominant haplotype followed by Hap03, Hap15, Hap06, and Hap10. Hap01 was found at all sampling sites and was considered the ancestral haplotype. A network with an ancestral haplotype typically shows a star-like or star-burst appearance with the ancestral haplotype centered in it (Ferreri, Qu & Han, 2011).

Figure 2 Maximun likelihood (ML) gene trees show the relationship of Thunnus tonggol haplotypes inferred from (A) D-loop marker (tree was compressed for a better illusration) (B) ND5 gene.

Branches were drawn to scale and bootstrap values < 50% were not shown. (The original D-loop ML tree was presented in Supplemental 1). All regions = SCS-1, SCS-2, SOM, CS, WCIO, ECIO, TW; West coast Indian Ocean (WCIO).

Figure 3 Haplotypes network diagram inferred from D-loop region.

Node size corresponds to the haplotype frequencies; minimum node size is one individual. Black dot represents median vector. Dashed line represents nucleotide mutation. Populations: KP, Kuala Perlis; PR, Pantai Remis; SB, Sungai Besar; TG, Pulau Kambing; KT, Kuantan; TB, Tok Bali; KC, Kuching; BT, Bintulu; MR, Miri; KK, Kota Kinabalu; SM, Semporna.

Figure 4 Haplotype network diagram inferred from the ND5 gene.

Node size corresponds to the haplotype frequencies; minimum node size is one individual. Black dot represents the median vector. Dashed line represents a nucleotide mutation. Populations: KP, Kuala Perlis; PR, Pantai Remis; SB, Sungai Besar; TG, Pulau Kambing; KT, Kuantan; TB, Tok Bali; KC, Kuching; BT, Bintulu; MR, Miri; KK, Kota Kinabalu; SM, Semporna.

Pairwise comparison ФST analysis corroborated the low and non-significant population structure of T. tonggol from Malaysian waters (D-loop: −0.0324 to 0.1191 (Table 3); ND5: −0.0254 to 0.0739 (Table 4)), except for seven and a single significant pairwise comparison involving BT, based on the D-loop and ND5 sequences, respectively. Further genetic differentiation assessment based on HST (D-loop: 0.0033; ND5: 0.0034), NST (D-loop: 0.0113; ND5: 0.0062) and KST* (D-loop: 0.0064; ND5: 0.0049) produced low and not significant values, which corresponds with a high level of gene flow (Nm) among Malaysian T. tonggol populations (D-loop: 145.69 and ND5: 146.01 for haplotype-based statistic; D-loop: 43.72 and ND5: 102.46 for sequence-based statistic). Correspondingly, the pairwise genetic distances among populations also exhibit relatively low values ranging from 0.0197 to 0.0251 (D-loop) and 0.0020 to 0.0038 (ND5). The hierarchical AMOVA indicated that more than 99% of the total genetic variation in Malaysian T. tonggol was contributed by genetic differences within populations. Attempts to identify if population subdivisions exist among the T. tonggol populations (BT vs. other populations) returned a non-significant FCT value with less than 1% contribution to the total genetic variation, while more than 99% of the total genetic variation was contributed within populations, based on both datasets. The Mantel Test also supported earlier findings, demonstrating no correlation between genetic differentiation (pairwise ФST value) and geographical distance (D-loop: r = −0.3763, P = 0.08 and ND5: r = −0.0050, P = 0.43) among Malaysian populations.

Table 3 Pairwise ФST estimates (below diagonal) and genetic distance (upper diagonal) between sampling sites of Thunnus tonggol inferred by mtDNA D-loop region.

Region	Population	SOM	SCS-1	SCS-2	CS	ECIO	WCIO	TW	
	KP	PR	SB	TG	KT	TB	KC	BT	MR	KK	SM	
SOM	KP		0.0251	0.0245	0.0217	0.0218	0.0238	0.0230	0.0246	0.0241	0.0235	0.0234	0.0264	0.0366	0.0214	
PR	0.0070		0.0246	0.0225	0.0227	0.0247	0.0244	0.0249	0.0248	0.0238	0.0237	0.0273	0.0371	0.0219	
SB	−0.0028	−0.0212		0.0220	0.0221	0.0240	0.0236	0.0244	0.0240	0.0230	0.0230	0.0265	0.0360	0.0212	
SCS-1	TG	−0.0131	−0.0046	−0.0081		0.0197	0.0212	0.0211	0.0208	0.0211	0.0205	0.0204	0.0238	0.0344	0.0181	
KT	−0.0245	−0.0066	−0.0140	0.0007		0.0212	0.0206	0.0229	0.0220	0.0209	0.0210	0.0244	0.0349	0.0192	
TB	−0.0041	0.0104	−0.0012	−0.0118	−0.0249		0.0225	0.0239	0.0233	0.0225	0.0226	0.0261	0.0364	0.0208	
SCS-2	KC	−0.0270	0.0127	−0.0028	0.0024	−0.0324	−0.0225		0.0238	0.0233	0.0228	0.0224	0.0255	0.0358	0.0214	
BT	0.0873	0.0742	0.0715	0.0355	0.1191	0.0828	0.0961		0.0227	0.0230	0.0225	0.0259	0.0368	0.0204	
MR	−0.0033	0.0019	−0.0144	−0.0321	−0.0007	−0.0141	0.0028	0.0188		0.0224	0.0225	0.0258	0.0362	0.0207	
KK	0.0313	0.0192	0.0038	0.0067	0.0144	0.0110	0.0433	0.0915	−0.0053		0.0216	0.0252	0.0352	0.0191	
CS	SM	0.0088	0.0005	−0.0146	−0.0143	0.0062	−0.0006	0.0097	0.0546	−0.0172	0.0032		0.0255	0.0352	0.0197	
ECIO	0.1371	0.1488	0.1371	0.1512	0.1617	0.1506	0.1478	0.1998	0.1286	0.1630	0.1596		0.0350	0.0218	
WCIO	0.0663	0.0683	0.0489	0.0662	0.0662	0.0730	0.0622	0.1100	0.0640	0.0687	0.0656	0.0507		0.0328	
TW	−0.0222	−0.0284	−0.0384	−0.0476	0.0029	−0.0197	0.0286	0.0366	−0.0448	−0.0439	−0.0332	0.0741	−0.0247		
Notes:

Strait of Malacca (SOM): KP, Kuala Perlis; PR, Pantai Remis; SB, Sungai Besar; South China Sea 1 (SCS-1): TG, Pulau Kambing; KT, Kuantan; TB, Tok Bali; South China Sea 2 (SCS-2): KC, Kuching; BT, Bintulu; MR, Miri; KK, Kota Kinabalu; Celebes Sea (CS): SM, Semporna; East coast of Indian Ocean (ECIO); West coast of India Ocean (WCIO); TW, Taiwan.

Bold numbers indicate statistically significant after FDR correction at α = 0.05.

Table 4 Pairwise ФST estimates (below diagonal) and genetic distance (upper diagonal) between sampling sites of Thunnus tonggol inferred by mtDNA ND5 gene.

Region	Population	SOM	SCS-1	SCS-2	CS	
KP	PR	SB	TG	KT	TB	KC	BT	MR	KK	SM	
SOM	KP		0.0030	0.0023	0.0026	0.0021	0.0020	0.0025	0.0029	0.0026	0.0024	0.0025	
PR	−0.0075		0.0031	0.0035	0.0030	0.0028	0.0033	0.0038	0.0035	0.0032	0.0033	
SB	−0.0089	−0.0090		0.0027	0.0022	0.0020	0.0026	0.0030	0.0027	0.0024	0.0025	
SCS-1	TG	0.0082	0.0207	0.0283		0.0025	0.0024	0.0029	0.0031	0.0030	0.0028	0.0028	
KT	−0.0171	0.0046	−0.0031	−0.0055		0.0019	0.0024	0.0029	0.0026	0.0023	0.0024	
TB	−0.0037	0.0090	−0.0106	0.0076	−0.0150		0.0023	0.0027	0.0024	0.0022	0.0022	
SCS-2	KC	0.0058	−0.0069	−0.0074	0.0006	−0.0079	−0.0050		0.0032	0.0029	0.0027	0.0027	
BT	0.0425	0.0436	0.0739	−0.0077	0.0550	0.0460	0.0444		0.0032	0.0031	0.0031	
MR	−0.0077	−0.0011	0.0015	−0.0169	−0.0064	−0.0095	−0.0200	−0.0015		0.0028	0.0028	
KK	−0.0169	−0.0030	−0.0095	0.0133	−0.0203	−0.0067	0.0032	0.0539	−0.0023		0.0026	
CS	SM	−0.0064	0.0028	−0.0009	−0.0144	−0.0213	−0.0090	−0.0254	0.0235	−0.0242	−0.0106		
Notes:

Strait of Malacca (SOM): KP, Kuala Perlis; PR, Pantai Remis; SB, Sungai Besar; South China Sea 1 (SCS-1): TG, Pulau Kambing; KT, Kuantan; TB, Tok Bali; South China Sea 2 (SCS-2): KC, Kuching; BT, Bintulu; MR, Miri; KK, Kota Kinabalu; Celebes Sea (CS): SM, Semporna.

Bold number indicates statistically significant after FDR correction at α = 0.05.

D-loop sequences of T. tonggol from Taiwan (TW), the east coast of the Indian Ocean (ECIO), and some of the west coast of the Indian Ocean (WCIO) were clustered with the Malaysian haplotypes, while some other WCIO haplotypes were placed into another clade with high bootstrap support (Fig. 2A; Supplemental 1). ML tree partitioning was partly in agreement with the pairwise comparisons ФST, where TW was not significantly structured for Malaysia, ECIO nor WCIO. In contrast, all pairwise comparisons involving ECIO and WCIO against Malaysian’s populations were statistically significant after FDR correction at α = 0.05, except for WCIO-KT (Table 3). Meanwhile, WCIO and ECIO were not genetically subdivided from each other (P > 0.05) (Table 3). A hierarchical AMOVA revealed the existence of genetic subdivision between the Indian Ocean and Malaysian waters (FCT: 0.09, P < 0.05), yet 90.01% of the genetic variation within the Indo-Pacific region was contributed by genetic differences within populations. Pairwise genetic distances between TW and Malaysian populations ranged from 0.0181 to 0.0219, while the genetic distances were relatively higher for pairwise comparisons involving ECIO and WCIO, that is, from 0.0238 to 0.0371 (Table 3).

Demographic history

Negative values of Tajima’s D and Fu’s FS (all significant at P < 0.05) neutrality tests were detected in all populations inferred from both the mtDNA D-loop and ND5 gene (Table 2). Large differences in population sizes before (θ0) and after expansion (θ1) were detected, i.e (on average) 0.116 and 54,915.300 based on D-loop sequences, while 0.103 and 81,838.400 were based on the ND5 gene marker (Table 2). Corresponding to the τ value of 2.321 (ND5) and 8.514 (D-loop) (Table 2), the calculated expansion time for T. tonggol in Malaysian waters was 67,865 and 284,252 years ago inferred by ND5 and D-loop markers, respectively Bayesian skyline plot (BSP) analysis revealed two significant increases in effective population size that occurred 200,000 and 950,000 years ago based on the D-loop marker (Fig. 5A), while continuous expansion started 150,000 years ago with a more recent expansion around 100,000 years ago based on the ND5 gene marker (Fig. 5B). Goodness of fit tests (Hri and SSD) exhibited non-significant values for the overall samples (P > 0.05) (Table 2). Population demographic analysis of T. tonggol matched a unimodal distribution for overall samples (Fig. 6).

Figure 5 Bayesian Skyline Plots of the mtDNA (A) D-loop marker and (B) ND5 gene of Thunnus tonggol populations in Malaysia.

The Y-axis indicates effective population size (Ne) × generation time, while the X-axis indicates mean time in thousands of years before present. The thick line represents the average and the blue band represents the standard error.

Figure 6 Mismatch distributions (pairwise number of differences) for the mtDNA (A) D-loop region (B) ND5 gene of Thunnus tonggol showing the expected and observed pairwise differences between sequences with the respective frequencies.

Discussion

Genetic diversity

According to Grant & Bowen (1998), the past demographic history of populations can be clarified based on their contemporary haplotype diversity (H) and nucleotide diversity (π). These two sensitive indices are the basis of genetic diversity estimation of a population (Nei & Li, 1979). In this study, all populations of T. tonggol showed high haplotype diversity (D-loop: 0.990–1.000; ND5: 0.848–0.965) but low to moderate nucleotide diversity (D-loop: 0.0195–0.0250; ND5: 0.0017–0.0039) (Table 2). High haplotype diversity, coupled with low nucleotide diversity, indicates a large population size that has undergone recent population expansion, which allows the retention of new alleles in the population but without sufficient time for accumulation of more nucleotide substitutions among haplotypes (Grant & Bowen, 1998; Chen et al., 2015; Delrieu-Trottin et al., 2017; Nabilsyafiq et al., 2019). These results were in agreement with previous findings of several pelagic fishes, including the spotted mackerel, Scomber australasicus (H = 0.996, π = 0.007) (Tzeng, 2007), yellowfin tuna T. albacares (H = 0.997, π = 0.035), skipjack tuna Katsuwonus pelamis (H = 0.999, π = 0.084) (Ely et al., 2005), and longtail tuna, T. tonggol (H = 0.999, π = 0.0016) (Willette, Santos & Leadbitter, 2016). Furthermore, the wide difference in nucleotide variability estimates between the two markers was consistent with the findings of Viñas & Tudela (2009), where the D-loop region had a ten-fold higher value of nucleotide diversity compared to the mtDNA ND5 gene.

Phylogenetic and population level analyses

Populations of T. tonggol from Malaysian waters exhibited an absence of geographical structure associated with mtDNA sequences, as evidenced by the single-clade gene trees (Fig. 2; Supplemental 1), ambiguous genetic partitioning of haplotype networks (Figs. 3 and 4), low and non-significant values of pairwise ФST (Tables 3 and 4) (except for several comparisons involving BT population), high contribution of within population variation through AMOVA, and non-significant correlation between genetic differentiation and geographical distance. These results strongly suggest that the T. tonggol populations in Malaysian waters were panmictic with shallow genetic structure due to high gene flow, similarly reported in other studies of the same species (Kunal et al., 2014; Willette, Santos & Leadbitter, 2016) and several other species (Carpenter et al., 2011). Wright (1931) suggested that the level of genetic differentiation among populations is related to the rate of evolutionary processes, like migration, mutation, and drift. Thus, a highly migratory species with a large population size, such as T. tonggol, is predicted to show limited population partitioning. BT population showed significant genetic structure from the rest (except TG and MR) and SB, inferred from the D-loop and ND5 sequences respectively, based on the population pairwise ФST analysis, however, all other analyses suggested genetic homogeneity with other T. tonggol populations in Malaysian waters. We believe that this could be due the different weightage of algorithms or characters used in the various analyses, perhaps a different emphasis on nucleotide versus haplotype diversity.

Pelagic fish in the marine realm are well-known to exhibit little genetic divergence (Ely et al., 2005). The weak genetic structure observed in T. tonggol within the pelagic environment is typical of pelagic fish due to their biological and life histories (Fauvelot & Borsa, 2011; Pedrosa-Gerasmio, Agmata & Santos, 2014). T. tonggol spawns during the monsoon season (Koya et al., 2018) where the ocean circulation shift (upwelling and down welling) during this season would enrich the water at the surface and thus lead to the growth of plankton (Yohannan & Abdurrahiman, 1998). Although T. tonggol is not a plankton feeder, the plankton bloom somehow enriches the food resources for other fish that become the prey of T. tonggol, hence, creating an optimal spawning ground for the species. T. tonggol is believed to spawn close to coastal waters (Nishikawa & Ueyanagi, 1991), thus the dynamic movement of waters during monsoon, not only helps in circulation of rich nutrients, but also in the larvae dispersal that could span a larger area (Madhavi & Lakshmi, 2012). Another possible explanation for the absence of limited gene flow among populations is the pattern of migration in the adult stage. In addition to the high dispersal potential during egg and larval stages, adults are characterized by high maneuverability during seasonal migration.

An earlier study by Willette, Santos & Leadbitter (2016) showed delineation of the Indian Ocean versus the South China Sea (samples were collected from Vietnam, Indonesia, and the Philippines) but without representatives from Malaysian waters. In the present study, based on the increased sample populations on a finer scale within this biogeographical region and GenBank sequences, we hypothesize that the genetic barrier lies within the Andaman Sea, which hindered partial gene flow between the coast of India and Malaysian waters, as evidenced in the ML tree (Fig. 2A; Supplemental 1), pairwise comparisons ФST (Table 3), and hierarchical AMOVA. There was also another possible break between ECIO and WCIO based on results in the ML tree (Fig. 2A; Supplemental 1) and pairwise comparisons ФST, though the moderate pairwise ФST value is not significant (Table 3). However, the phylogenetic relationship of T. tonggol from Malaysian waters and other regions of the South China Sea remains unknown due to the limited genetic data and unavailability of the haplotype sequences from the study by Willette, Santos & Leadbitter (2016) in the public database. We postulated that close genetic relationships would be expected, based on the recent findings regarding the absence of genetic subdivision between TW and Malaysian T. tonggol. Future studies should include more detailed sampling within the Andaman Sea and adjacent waters to substantiate this.

Demographic history

Historical events during the Pleistocene epoch could have shaped the genetic diversification of T. tonggol populations observed in the present study. All relevant statistical tests implied a scenario of past population/demographic expansion in the absence of background selection. Negative Fu’s FS values signified the alterations caused by population expansion and/or selection (Fu, 1997), which was further supported by Tajima’s D that implied a notable population growth or genetic hitchhiking in a background of recent excess mutations (Tajima, 1989). Likewise, the non-significant sum of squared deviations (SSD) and Harpending’s raggedness index (Hri) indicated the occurrence of population expansion in T. tonggol (Kunal et al., 2014) that inhabits Malaysian waters. Furthermore, the star-like pattern of the median–joining network (Fig. 4) and unimodal pattern of mismatch distribution (Fig. 6) further support the occurrence of a sudden demographic expansion during recent history of the taxa (Slatkin & Hudson, 1991; Rogers & Harpending, 1992; Ferreri, Qu & Han, 2011; Kunal et al., 2014; Chen et al., 2015; Pedrosa-Gerasmio, Agmata & Santos, 2014).

The large population size differences before (θ0) and after expansion (θ1) also suggested a rapid population expansion of T. tonggol in the past as also reported in a previous study of populations from India (Kunal et al., 2014). The overall τ value observed in Malaysia was much lower than T. tonggol populations from Indian waters, which was 21.26. The estimated time for population expansion in Indian waters was 593,334 years before the present (Kunal et al., 2014), as compared to 67,865 and 284,252 years ago inferred by the ND5 and D-loop markers, respectively, for T. tonggol from Malaysian waters. This suggests that T. tonggol in the Indian region underwent earlier expansion, with subsequent large population retention. In this study, the ND5 gene marker was able to detect a more recent population expansion of T. tonggol populations in Malaysian waters around 70,000 years ago (based on tau value) and 100,000 years ago (based on BSP analysis). Based on the D-loop marker, two expansion events were detected, where the first round occurred around 200,000 years ago (based on BSP analysis) or 284,000 years ago (based on tau value) (during middle Pleistocene (126,000–781,000 years before present) (Saul, 2016)) and the second round occurred around 950,000 years ago (based on BSP analysis) (during early Pleistocene) (Fig. 5). In general, both markers were able to detect population expansion that occurred around 200,000 years ago (ND5 gene marker indicated population expansion started before 150,000 years ago).

Implication for fisheries management

To date, there is limited information on the population structure of T. tonggol, especially in Malaysia, and from the management point of view, this is a critical issue. The present study provides the first baseline population genetic data on T. tonggol populations in Malaysian waters, which is important information for management planning by authorities.

Managing fishery resources takes a significant effort to protect and replenish the genetic pools for a sustainable harvest. Molecular analyses, in complement with other approaches, may serve as a reliable measurement for an efficient preservation strategy (BjØrnstad & Ried, 2002; Toro, Barragan & Ovilo, 2003). The genetic data suggest that T. tonggol in Malaysia forms a panmictic population as observed by the wide distributional range of this species and non-significant low ФST values among the populations studied, thus suggesting a single evolutionary significant unit (ESU). However, this study was based on mitochondrial markers and therefore, restricted to only the pattern of maternal inheritance. For a holistic genetic perspective of bi-parental inheritance, co-dominant markers, such as microsatellites, should be included.

Conclusions

The T. tonggol populations in Malaysian waters revealed the absence of population structure as inferred by both mtDNA markers and therefore, could be regarded as a single stock unit for management purposes based on the current data. Their inferred demographic history suggests that T. tonggol populations expanded significantly during the middle and early Pleistocene. Overall, this study is a critical first baseline, providing insights for stock management of this neritic species in Malaysian coastal areas. Coupled with other related information, the assimilation of this genetic information could aid the development of effective management plans in the future, not only in Malaysia but also in neighboring countries sharing the same waters. Finally, this has contributed further insights into genetic locality, delineating the species within the Indo-Pacific biogeographical region.

Supplemental Information

Supplemental Information 1 Unscaled ML tree showing the relationship of Thunnus tonggol haplotypes inferred from the D-loop marker.

Hap001-042 (SCS-1), 043-085 (SOM), 086-104 (CS), 108-152 (SCS-2), 153-166 (ECIO), 168-296 (WCIO), 297-300 (TW).

Click here for additional data file.

We highly appreciate assistance by the officers and crew of the research vessel MFRDMD/SEAFDEC and Department of Fisheries (DoF) Malaysia for sharing their expertise and knowledge and also facilitating sample collections. We are also grateful to our laboratory colleagues for their technical assistance. We are forever grateful to Dr. Adelyna Mohd Akib and her PhD student, Mr. Danial Hariz for their valuable guidance and comment regarding Bayesian Skyline Plot analyses.

Additional Information and Declarations

Competing Interests

Author Contributions

Animal Ethics

Data Availability

The authors declare there are no competing interests.

Noorhani Syahida Kasim conceived and designed the experiments, performed the experiments, analyzed the data, prepared figures and/or tables, authored or reviewed drafts of the paper, and approved the final draft.

Tun Nurul Aimi Mat Jaafar conceived and designed the experiments, performed the experiments, analyzed the data, authored or reviewed drafts of the paper, and approved the final draft.

Rumeaida Mat Piah conceived and designed the experiments, authored or reviewed drafts of the paper, and approved the final draft.

Wahidah Mohd Arshaad conceived and designed the experiments, performed the experiments, analyzed the data, authored or reviewed drafts of the paper, and approved the final draft.

Siti Azizah Mohd Nor conceived and designed the experiments, analyzed the data, authored or reviewed drafts of the paper, and approved the final draft.

Ahasan Habib analyzed the data, prepared figures and/or tables, authored or reviewed drafts of the paper, and approved the final draft.

Mazlan Abd. Ghaffar conceived and designed the experiments, authored or reviewed drafts of the paper, and approved the final draft.

Yeong Yik Sung analyzed the data, authored or reviewed drafts of the paper, and approved the final draft.

Muhd Danish-Daniel analyzed the data, authored or reviewed drafts of the paper, and approved the final draft.

Min Pau Tan conceived and designed the experiments, performed the experiments, analyzed the data, prepared figures and/or tables, authored or reviewed drafts of the paper, and approved the final draft.

The following information was supplied relating to ethical approvals (i.e., approving body and any reference numbers):

No field permit or ethics approval were needed since we collected dead fish samples from the wet market.

The following information was supplied regarding data availability:

All haplotypes are available at GenBank: MK643829–MK644008 (D-loop) and MN252922–MN252981 (ND5).

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
