# Peer review of "Recent population expansion of longtail tuna Thunnus tonggol (Bleeker, 1851) inferred from the mitochondrial DNA markers"

_PeerJ, doi:10.7717/peerj.9679_

## Round 0.1 · original submission · Major Revisions

All three reviewers' raised a lot of questions, queries and also make several comments, especially regarding single mtDNA gene marker rather than a multigenes approach to produce more robust results. Also, the reviewers' suggested more comprehensive analysis like Bayesain analysis (BSP) to infer the time of expansion! Finally, if you are not able to address all the queries; and don't do robust analysis including more genes, the possibility of acceptance will be very slim!

Reviewer 1 ·

Basic reporting

MS entitled "Recent population expansion of longtail tuna Thunnus tonggol (Bleeker, 1851) inferred from the mitochondrial DNA D-loop marker" tries to identify the population structure of Thunnus tonggol in Malaysian waters . The methodology followed is fine and the work needs reporting. However some more concrete analysis like Bayesian analysis need to be done to determine the time of expansion of the population, rather than making hypothetical assumptions as in Line 263-274.

Experimental design

Experimental designs meet the general standards, though the article don't try to address the spatial and temporal variations.

Validity of the findings

Conclusions are well stated, linked to original research question.

MS concludes saying that the population is panmictic. However, Pairwise Fst values, show they are significantly different at least in 22 cases (See table 3).Author may justify this, before coming to a blanket conclusion like " panmictic".

Also improve the discussion, by considering the significant difference in pairwise Fst of populations.

Additional comments

More concrete analysis like Bayesian analysis must be be done to determine the time of expansion of the population, rather than making hypothetical assumptions as in Line 263-274.

Significant difference in pairwise Fst should be taken care and reexamine if there is any fine scale structuring among populations?

Result of AMOVA Fst and its significant level should be given in result and explained

Reviewer 2 ·

Basic reporting

The manuscript was well structured, however, some statements are not clear and the overall English used can be improved.

Some sentences may make reader confuse and some terms used were inappropriate.

Lines 60-61: What does "unique pattern of distribution" means? What is the relevant of distribution pattern with standing in tuna fisheries globally?

Lines 67-68: The catches is increase or decrease? (291264 tonnes in 2007; 237124 tonnes in 2016)

Line 104: What does “to establishing baseline data and stock delineation pattern” means?

Line145: interpreted?

Line 169: investigate relationship or correlation?

Line188: obtained from?

Line 228: explored or assessed?


Literature review is sufficient, however, some explanations are not clear.

Line 75: This statement is not relevant in this study.

Lines 77-87: What do authors would like to express in this paragraph by mentioning different Thunnus species population studies?

Line 91: to describe population…Regarding what?

Lines 92-95: Should rewrite this statement because not all mtDNA region are highly variable. It is also contained some conserve region as well.

Line 96: Inappropriate reference was used (Fliss et al., 2000). Please look for the mutation rate for fish not human. Mutation rate mtDNA in fish is not up to 10-17 folds higher than nuclear DNA.

Line 102: Why need to include phylogenetic relationship of T. tonggol from Taiwan waters and the northwest coast of India in this study? How this phylogenetic relationship will relate to the present study?


Tables 2 and 3:
1) What is the purpose to include the ϴs value but no explanation or discussion about it?
2) Please standardize the decimal number of values. e.g. 1 ->1.000; 0.25 -> 0.0250

Figure 1:
1) Should include the longitudes and latitudes.
2) Location of India and Taiwan population were not plotted in the map.

References:
Some references were not cited in text: lines 339, 342, 397 and 438.

Experimental design

No issue for ethical statement and field work permit. The English writing for this statement can be improved.

Identification, preservation and storage of specimen were not explained.

Analyses solely based on single mtDNA gene marker will provide limited information. More gene markers should be included to product more robust result. Single gene marker is unable to reflect the true phylogenetic relationship among the samples. In present study, the constructed tree is considered as gene tree not phylogenetic tree. Please use the term correctly. The constructed tree also should be provided to enable reader to confirm the results explained by the authors.

Additional Minimum Spanning Network construction should be conducted in which it can better explain the relationship among the haplotypes compare to the gene tree.

AMOVA result also need to be presented such as FCT value, population/group partitions and etc..

Other comments:

Line 126: fin or fin tissue?

Line 127: PCR should be written in full sentence.

Line 131: No description about Mytaq. Is that a Premixed solution?

Line 134: The purpose to visualize on agarose gel not explain.

Line 138: This paragraph explains about data editing/verification, not data analyses.

Lines 150-154: Refer comment for line 102 and experimental design.

Lines 166-167: Genetic divergence within population and between haplotypes was calculated respectively using MEGA 6. (Results not shown)

Line 171: What is linear distance?

Validity of the findings

As mentioned, information provided by single gene markers are limited and not robust. Additional analyses as suggested is needed.

Interpretation of results not clear and still need to be improved.

Explanation of some results and discussions are confused and will pointed out below.

Lines 190-191: The majorities (92.2%) of haplotypes are private to single locality indicating some degrees of population structured or lack of gene flow. This is contradicts to the panmictic population as claimed by the authors. Network analysis may help to clarify this issue.

Lines 199, 202: The constructed tree is need to be presented.

Line 198: What does “single cluster with unsupported clades (< 90%)” means?

Line: 203: AMOVA result is need to be presented? How about the FCT value?

Line 208: According to Table 3, some comparisons (especially comparison of BT with others [>0.05]; TG, PR..) show significant moderate FST value, which indicating existence of population structuring. Why authors claim low differentiation? Refer the comment for lines 190-191 also. Panmictic population seems like inappropriate.

Line 211: Does the average of distance 1217.6km means anything in this analysis?

Line 219: The statement is not correct based on the above comments.

Line 230: All of the nucleotide diversity values are around 0.02; ranged from low to moderate was not observed.

Line 242: Refer the comment for line 219.

Line 249: It is not uncommon that Taiwan is closely related to Malaysia populations as they are located at South China Sea as well and distinct from India's population. However, the population from west Malaysia (KP, PR and SB) are considered part of Indian Ocean. So, if these population were mixed with the South China Sea populations, then the suggested genetic partition is seens not related to the Sunda Shelf barrier.

Line 264-274: The authors try to explain and relate the demographic pattern with the paleogeographical changes of Indo-Malay region. However, the explanation was not clear/well. What is the consequence to the demography changes when the limited population distribution occurred during lower sea level? How the submergence of Sunda shelf have fueled the population expansion?
Population expansion time estimation may clarify whether the demography changes was related to the cyclical glaciation event or not.

Line 281: The statement “deemed as a single stock unit and was found to be highly diversified” is contradicts.

Line 282: Based on what results the authors claimed that “the species’ vulnerability to extinctions may be low”?

The overall conclusion are not well stated.

Additional comments

There is still room for improvement especially in data analyses and interpretation.

Reviewer 3 ·

Basic reporting

This work is a valuable effort for improving the knowledge on the highly exploited species.
The amount of work to produce and analyze the dataset is evident.
I consider this manuscript properly structured and presented, but the data analysis should be improved and integrated with multiple and recent analytical methods to have comprehensive and robust results to be assessed and discussed.
The dataset is constituted by a single mtDNA marker and this limitation should be accounted for and results should be discussed with consequent awareness.

Experimental design

This work is a valuable effort for improving the knowledge on the highly exploited species.
The amount of work to produce and analyze the dataset is evident.
I consider this manuscript properly structured and presented, but the data analysis should be improved and integrated with multiple and recent analytical methods to have comprehensive and robust results to be assessed and discussed.
The dataset is constituted by a single mtDNA marker and this limitation should be accounted for and results should be discussed with consequent awareness.

Validity of the findings

This work is a valuable effort for improving the knowledge on the highly exploited species.
The amount of work to produce and analyze the dataset is evident.
I consider this manuscript properly structured and presented, but the data analysis should be improved and integrated with multiple and recent analytical methods to have comprehensive and robust results to be assessed and discussed.
The dataset is constituted by a single mtDNA marker and this limitation should be accounted for and results should be discussed with consequent awareness.

Additional comments

I attach a pdf file where you will find more punctual remarks inserted as comments.

Annotated reviews are not available for download in order to protect the identity of reviewers who chose to remain anonymous.

---

## Round 0.2 · Major Revisions

Still there are many things to consider! check carefully the reviewer comments, especially "The manuscript seems like not ready for submission as there are still a lot of mistake, not well-prepared data (table’s caption, decimal number, low resolution of figure), weak data explanations (will be pointed out below) and not organised references. Please check in detail before submitting.

Reviewer 2 ·

Basic reporting

The manuscript seems like not ready for submission as there are still a lot of mistake, not well-prepared data (table’s caption, decimal number, low resolution of figure), weak data explanations (will be pointed out below) and not organised references. Please check in detail before submitting.

Literature review is sufficient. However, some sentences may make reader confuse and some terms used were inappropriate.

Line 66: author claimed, “decrease in its catch from 291,264 tonnes to 67 237,124 tonnes (FAO, 2018b) from 2007 to 2016” but in line 78 author claimed, “steady increases in fisheries catches over the years”. So, what author try to explain? Increasing in demand, I think.

Line 98: Inappropriate reference was used (Haag-Liautard et al., 2008). This reference is about mutation rate in Drosophila melanogaster. Please look for the mutation rate for fish.

Line 103: The term “COMPREHENSIVE” doesn’t sound suitable without including nuclear gene markers in this study.

Lines 105-111: It is not necessary to include this explanation.

Lines 116-121: This sentence need to be rewritten to make the statement clear.

Lines 122-129: Based on my understanding, the main objective of this study is to conduct population study of T. tonggol in Malaysian water and further investigate the phylogeography of T. tonggol in Indo-Pacific region. So, please rewrite this paragraph to make the objectives of this study clearer.

Line 126: Use the term “Provided by…” instead of “contributed by…”; “further investigate” instead of “expand the analysis”.

Experimental design

No issue for ethical statement and field work permit.

Research question is relevant and meaningful. However there are room to be improved in data analyses.

Line 138: “sample sizes ranged from 13 to 23” but based on Table 2, the sample size ranged from 12 to 24. And the average of 18.9 individuals per sampling location is meaningless as you can not get 18.9 individual.

Line 139: Following the identification keys as described in Lim et al., 2018, not “using fish identification books”.

Line 156: Genomics DNA extraction or isolation is more appropriate in this context.

Line 167: Any reason why 32 cycles of amplification and final hold at 4oC for D-loop while 35 cycles and final hold at 10 oC in ND5?

Line 172: Are you sure the main purpose to visualized on agarose gels is to estimate the size of DNA fragment?

Line 176: Please don’t use symbol (&) in sub-title.

Line 198-202: What does the author try to explain? Inclusion of Taiwan and India sequences in phylogenetic tree or haplotype network?

Line 199: “courtesy of researchers…”?

Line 203: Why use Tamura-Nei model?

Line 238: Why use GTR model? Why use different model for same dataset in different analyses (tree construction, Pairwaise FST and Bayesian analyses)?

Line 239: The analysis was run for 108 iterations with a pre-burn-in of 107. Are you sure? After burn-in only left 10 runs?

Please use suitable term in explanation e.g.:
Line 186: “inspected”?
Line 196: “to view”?
Line 197: “phylogenetic network”?
and much more in the whole manuscript.

Validity of the findings

Weak data interpretation.

Line 267: Table2: label (location) for first data was missing. Decimal number not consistent for all data.

Lines 273 and 289: 1) Are you sure these are the unrooted trees?
2) What is the purpose to include the outgroup in the constructed tree?
3) Why there is no bootstrap value? I’m doubt that there is no single clade having high bootstrap value.

Lines 277 and 299: 1)The constructed tree (Fig. 3) didn’t not shows that the T. tonggol from Adaman and India differentiated from the rest as the haplotypes 55, 56, 59, 4 and 8 from this region are clustered together with the others in the main clade which is polytomy. Only two haplotypes (9and 11) were in different clade and no strong bootstrap value to support.
2) Many haplotypes were missing/ not included in the tree.
3) What do you mean “All positions with less than 50% site coverage were eliminated”?
4) “Molecular Phylogenetic analysis” is not suitable as only one gene was used.

Lines 280, 283 and 299: Fig 4A is haplotype network for D-loop or ND5, same for Fig 4B. The resolution of the figure is too low until the label can’t to be read.

Line 316: Pairwise Fst value for Bintulu are consider moderate as there are more than 0.05. For tables 3 and 4, number of replications is 1000 or 10000 as stated in materials and methods? The decimal number should be consistent for all data.

Line 320: Any explanation for Table 6?

Lines 355-357: Which result that you referred? Pairwise Fst or genetic distance? For pairwise Fst, it tells the gene flow between populations or population structuring but not the genetic differentiation between population. Genetic differentiation between sequences from Taiwan and Malaysia is around 2% (not lack of genetic differentiation). Please make sure you know how to interpret the pairwise Fst and genetic distance before making the statement.

A lot of the pairwise Fst value and genetic distance for Malaysia populations in Table 7 are different from Table 3. Please check.

Line 373: Based on Table 2, all Tajima’s D value for D-loop are not significant. Any explanation?

Lines 379 and 384: Based on the Bayesian skyline plot, the expansion time occurred at around 3(not 5) kya for D-loop and 2kya for ND5. Why author suggest that “a recent expansion that could reasonably have followed the cyclic climatic oscillations defined by the late Pleistocene era (~110 – 15 kya)” which was happened more earlier? (expansion time not in the Pleistocene time range).

Lines 389: Any explanation for the huge differentiation of the expansion time between Bayesian result and the calculation using tau value (5 kya; 2kya vs 55kya;323kya) although both analyses using same mutation rate. Which result are more reliable?

Lines 402-403: Fig 4A illustrated complex reticulation while Fig 4B illustrated star-like pattern. Please make sure Fig 4A is D-loop or ND5 and also for Fig4B.

Lines 423-424: This statement is not clear.

Line 434: What does “high haplotypes” means?

Lines 451-452: Any other better explanation for population structuring of BT population as both gene markers reveals nearly same result.

Line 466: This statement is contradicting to the discussion of this paragraph.

Line 471: Refer the comment for lines 379 and 384.

Lines 494-506: In the first sentence, author want to relate the changes in geological features and climate event in the past with the demographic pattern observed. However, further discussion does not relate the changes of sea level with population demographic pattern but relate to the population structuring. It is confusing.

Lines 503-506: It is farfetched by relating the short period of LGM with low population structuring. Low population structuring should be related to the ocean currents, high dispersal ability of T. tonggol and the rising of sea level which promote gene flow as discussed in the other paragraphs.

Lines 510 -515: Based on the expansion time discussed in line 488, it was happened far more earlier than the LGM. So, the expansion of T.tonggol in the present study seems not related to the rising of sea level after LGM.

Lines 519 -512: The explanation of the hypothesis was not clear. What does “intermediate and parallel genetic distance” means?

Additional comments

Week data interpretation.

Some fundamental knowledge should be improved such as how to differentiate rooted and unrooted tree and how to interpret tree; don’t confuse with genetic distances, pairwise Fst value, population structuring, population demography, etc…

Please check carefully and well prepared before submitting manuscript.

References
A lot of references are not organised and sorted accordingly.

---

## Round 0.3 · Minor Revisions

Thank you for improving the manuscript but still there are some minor issues to address. please do the needful carefully that will significantly increase the acceptability of the manuscript!

Reviewer 2 ·

Basic reporting

The manuscript was well structured; Literature review is sufficient, well referenced & relevant.

Figures are relevant and high quality, however, some mistakes were found.

Abstract:

Line 28: In this study, past demographic history (population expansion) may promote gene flow between populations but this is just deduced from the demography history’s result, however, not certainly. So, the term “attributed to” is not appropriate. Low genetic differentiation may cause by many other factors such as no barrier, high dispersal ability, recent gene flow, etc… (Refer to lines 419 -432). Please revise this statement.
Line 31: AMOVA is mainly for testing whether there is obvious different grouping/partition as author stated and not for testing gene flow among study sites. Pairwise FST value is more appropriate for this statement.

Introduction:

Line 58: The status of subgenus Neothunnus is ambiguous. Any references? Please check.

Experimental design

No issue for ethical statement and field work permit.

Research question is relevant and meaningful.

Should be more careful and detail in describing methodology.

Line 140: The sampling sites can be easily divided into 4 geographical regions. Is it necessary or any reason to follow Akib et al., 2015?

Line 141: Please state how many sequences from Indian water.

Line 193: Your model selection is based on the model test and the model availability of the selected program (e.g. Beast/Arlequin). If the best model not available, use second or third best model. Why follow Kunal et al., 2014 as their dataset is different from yours.

Line 197: “Malaysian haplotypes”? Please use proper term.

Line 200: Based on the explanation in lines192 and 193, I guess you are using weights genetic distance for pairwise FST (population comparison setting in Arlequin [compute distance matrix]). So, the use of the term “phi (Ф)ST” is more appropriate instead of FST.

Line 212: Make sure you are fully understood about FST (Wright’s fixation index). FST is describes variation between individuals within populations ?? Or FST describes differentiation between population. Don’t simply links the AMOVA result in Arlequin with the Wright’s fixation index. Please refer to Arlequin ver 3.5.2’s manual section 8.2.1.2.

Line 226: How you get 315 sequences in total as sequence from Genbank is 157 plus 152 from Malaysian haplotypes?

Line 238: The mutation rate of 2% per million years for ND5 is per site or per gene?

Lines 252- 254: Population in equilibrium will shows multimodal distribution, but multimodal not necessary indicate demographic equilibrium, it can be indicating other situations as well. Same for population with expansion will show unimodal……. Be careful in writing statement.

Validity of the findings

Data presentation and interpretation need more careful and to be improved.

Line 279: The phylogenetic gene tree in Fig 2A doesn’t match with the supplementary 1. Please state clearly different type of trees produced e.g. scaled tree or not scaled tree, etc…
- Hap 218 and Hap 231 is clustered in the same clade with “All region and Hap167, 253” in supp.1, but in Fig. 2A there are in polytomy status.
- Some branches with no taxa label.
- Species name Thunnus orientalis not Iltalic.

Line 279: Some taxa with no label and species name not Italic.

Lines 282-283: The Fig.3 is very messy and hard to “read”, should be re-arranged. The color version (previous submission) of haplotype network will be better. The length of the line should be proportion to the number of mutation steps. e.g. line with one mutation should be shorter than the line with more mutations.

Line 363: The overall tau value should based on the value generated by DnaSP or Arlequin by consider all samples as one single population dataset, and should not based on the mean value as shown in Table 2.

Line 364: Different gene markers show different estimated expansion time don’t mean the expansion time was happened from 65,438 to 323,217 years before present. Please revise the statement.

Lines 364-367: Any explanation for the huge differentiation of the expansion time between the calculation using tau value and Bayesian result (e.g. ND5; 65438 [based on tau value] vs 150000 [BSP]) although both analyses using same mutation rate. Which result are more reliable?

Lines 425-426: Even though plankton bloom will enrich the food resources, but why creating an optimal spawning ground for the species since T. tonggol is not plankton feeder? This statement is confusing.

Line 437: According to the Fig. 2A, Supplementary 1 and pairwise comparisons FST (Table 3), there are also another possible break between ECIO and WCIO though the moderate pairwise FST value is not significant.

Lines 444-448: You just mentioned other tuna species (Akib et al., 2015) also have same pattern, yet no explanation why different tuna species can have the same pattern. What are the factors that cause different species having same pattern?

Line 469: It doesn’t make sense that the population keep expanding from 65438 to 32217 years ago. Please refer to the comment for line 364.

Line 470: Which result indicates that Indian region with subsequent population retention?

Line 472: Refer to the comment for line 469.

Lines 472-475: You should have better explanation or more support information if you want to explain that the cyclical glaciation during Pleistocene have contributed to the population expansion but not just possibly because of increasing climatic conformity only.
Line 476: You can not simply state that LGP has no influence. Many occurrences can be happened during LGP that help in maintaining the large population size. You need to prove or explain this statement, not just make a statement in just one sentence.

Additional comments

Data presentation and interpretation need more careful and to be improved.

---

## Round 0.4 · Minor Revisions

After extensive review, the science is in good shape for publication, however, James Reimer, the Section Editor, has commented and said: "The English is not up to par; even in the Introduction this is soon clear, with even simple singular/plural verb problems and missing verbs."

PeerJ does not provide editing as a standard part of the review process but can offer it for a separate charge. Alternatively, the authors can consult with a fluent English-speaking colleague who is familiar with the subject matter or engage a professional scientific editing company. So, please take the necessary actions to make the English as PeerJ standard to be published!

Reviewer 2 ·

Basic reporting

This revision is almost ready for publication, however, a few issues need to be addressed especially on the population expansion time.

Experimental design

no comment

Validity of the findings

Figure 3: Please follow the editor team’s advice whether the color or pattern figures to be presented.

Supplementary 1: “Unscaled” ML tree shows……
The last sentence “Several haplotypes…..” Do you mean “Individuals for several haplotypes distributed in more than one regions.”?

Lines 356 and 462: It is not between 67,865 to 284,252 years. It should be 67,865 and 284,252 years ago respectively inferred by ND5 and D-loop genes.

Line 359 : For BSP, continuous expansion started from 150,000 years ago and more recent expansion around 100,000 years ago based ND5.

For expansion time, it can be summarized in this way:

- ND5 able to detect more recent expansion time around 70,000 years ago (based on tau value) and 100,000 (based on BSP).

-D-loop, 2 expansion events occurred:
1st: around 200,000 (BSP) or 284,000 years ago (tau value)
2nd: around 950,000 years ago (BSP)

In general, both markers able to detect population expansion occurred around 200,000 years ago (ND5, started before 150,000 years ago).

---

## Round 0.5 · accepted · Accept

First, I would like to say 'THANK YOU' to our honorable reviewers' for their outstanding effort for shaping the manuscript in a publishable form!
To authors: Congratulations! Finally!